# Longitudinal Trajectory Modeling to Assess Adherence to Sacubitril/Valsartan among Patients with Heart Failure

**DOI:** 10.3390/pharmaceutics15112568

**Published:** 2023-11-01

**Authors:** Sara Mucherino, Alexandra Lelia Dima, Enrico Coscioni, Maria Giovanna Vassallo, Valentina Orlando, Enrica Menditto

**Affiliations:** 1CIRFF, Center of Pharmacoeconomics and Drug Utilization Research, Department of Pharmacy, University of Naples Federico II, 80131 Naples, Italy; sara.mucherino@unina.it (S.M.); valentina.orlando@unina.it (V.O.); 2Health Technology Assessment in Primary Care and Mental Health (PRISMA), Institut de Recerca Sant Joan de Déu, Santa Rosa 39-57, 08950 Esplugues de Llobregat, Spain; alexandra.dima@sjd.es; 3Division of Cardiac Surgery, Azienda Ospedaliera Universitaria San Giovanni di Dio e Ruggi d’Aragona, 84131 Salerno, Italy; coscionienrico@gmail.com (E.C.); mgvassallo@hotmail.it (M.G.V.)

**Keywords:** medication adherence, heart failure, adherence trajectories, cluster analysis

## Abstract

Medication adherence in chronic conditions is a long-term process. Modeling longitudinal trajectories using routinely collected prescription data is a promising method for describing adherence patterns and identifying at-risk groups. The study aimed to characterize distinct long-term sacubitril/valsartan adherence trajectories and factors associated with them in patients with heart failure (HF). Subjects with incident HF starting sac/val in 2017–2018 were identified from the Campania Regional Database for Medication Consumption. We estimated patients’ continuous medication availability (CMA9; R package AdhereR) during a 12-month period. We selected groups with similar CMA9 trajectories (Calinski-Harabasz criterion; R package kml). We performed multinomial regression analysis, assessing the relationship between demographic and clinical factors and adherence trajectory groups. The cohort included 4455 subjects, 70% male. Group-based trajectory modeling identified four distinct adherence trajectories: high adherence (42.6% of subjects; CMA mean 0.91 ± 0.08), partial drop-off (19.6%; CMA 0.63 ± 0.13), moderate adherence (19.3%; CMA 0.54 ± 0.11), and low adherence (18.4%; CMA 0.17 ± 0.12). Polypharmacy was associated with partial drop-off adherence (OR 1.194, 95%CI 1.175–1.214), while the occurrence of ≥1 HF hospitalization (OR 1.165, 95%CI 1.151–1.179) or other hospitalizations (OR 1.481, 95%CI 1.459–1.503) were associated with low adherence. This study found that tailoring patient education, providing support, and ongoing monitoring can boost adherence within different groups, potentially improving health outcomes.

## 1. Introduction

Studies on people living with heart failure (HF) with reduced ejection fraction (HFrEF) widely demonstrated underutilization of HF drugs, failure to reach target doses, and poor adherence to treatments [1,2,3]. Low adherence to HF treatments has been shown to be the cause of increased mortality rates and hospital readmissions for patients with worsening disease [4]. Hence, The European Medicines Agency (EMA) in 2015 approved sacubitril/valsartan (sac/val) for the treatment of patients with symptomatic chronic HF with reduced ejection fraction [5]. Sac/val efficacy, compared with enalapril, an angiotensin-converting enzyme (ACE) inhibitor, was evaluated in the PARADIGM-HF randomized controlled trial [6,7,8,9], finding a significant reduction in re-hospitalizations for HF and risk of cardiovascular mortality compared with people living with HF treated with the ACE inhibitor. Albeit recorded levels of adherence to sac/val are defined as acceptable in several US studies [10,11,12], it still appears that early discontinuation of therapy is not uncommon. These treatment failures have reduced the potential population-level benefit that this therapy might otherwise offer.

It is now widely recognized that low medication adherence is a significant public health concern, particularly in chronic patients, and contributes to increased morbidity, mortality, and health care costs [13]. Attempts to quantify the degree to which subjects follow their medication regimen have resulted in the development of various adherence measures [14]. A recent study investigates on common pitfalls of adherence estimation using electronic health databases (EHDs), health-related archives created for administrative purposes and linked through patient identifiers to create comprehensive population databases, allowing the reconstruction of treatment profiles [14,15,16]. The study found that slight changes in definition can significantly skew medication adherence estimation from EHDs, potentially creating misleading insights on drug efficacy in real-world studies [14]. Efforts have been undertaken to alleviate the problems through a systematization of terminology and definitions, through the release of the EMERGE guidelines [17] and the ABC Taxonomy for Medication Adherence publication [18], as well as examples of medication adherence estimation from administrative databases [19,20]. Regarding the latter, the use of group-based trajectory modeling (GBTM) within the medication adherence literature is rapidly growing [21], and few studies to date have evaluated medication adherence from EHDs using the GBTM-longitudinal approach. Hence, the application of GBTM based on longitudinal adherence behavior allows to identify groups of subjects with a similar pattern of adherence behavior over a period of time, e.g., with similar adherence trajectories [12,22,23,24,25]. This study aimed to characterize distinct long-term sacubitril/valsartan adherence trajectories and the factors associated with them in subjects with heart failure [19].

## 2. Materials and Methods

### 2.1. Study Design, Population, and Data Sources

A retrospective observational study of subjects with heart failure initiated with sacubitril/valsartan (sac/val) treatment was carried out in Campania Region (Southern Italy). Sac/val (ATC Code C09DX04) is a complex of the neprilysin inhibitor prodrug, sacubitril, and the angiotensin receptor blocker (ARB), valsartan, which was recently approved in Europe and the United States for the treatment of chronic heart failure (HF) with reduced ejection fraction (HFrEF) and left ventricular ejection fraction (LVEF) ≤40% and appeared on the Italian market in May 2017. Sac/val is available in three different dosages: 24 mg sacubitril/26 mg valsartan (low dosage); 49 mg sacubitril/51 mg valsartan (medium dosage); and 97 mg sacubitril/103 mg valsartan (high dosage).

All subjects who received a prescription of sac/val between 1 May 2017 and 31 May 2018 were included. For each subject, an index date was identified, intended as the date of the first prescription of one of the three dosages of sacubitril/valsartan. Subjects who moved out of the region after the index date (as data might be missing) and subjects with less than one year of follow-up available were excluded. All subjects who reflected the conditions above described (flow chart in Appendix A) were characterized at the index date and were observed from the index date for 12 months until 31 May 2019 (end of the study period) or until the date of interruption of treatment or date of death (Figure 1).

The electronic health database used as a data source was a population health-related data warehouse generated from the data contained in the individual administrative databases, CaReDB, which includes data that have been validated in previous drug utilization studies [26,27,28,29,30,31,32]. The database contains information on the personal data of patients/doctors, pharmaceutical prescriptions, outpatient prescriptions, and hospital discharge forms of the individual LHUs of Campania Region, Southern Italy (about 6 million inhabitants). Databases were connected to each other through a record-linkage system that used as a key the identification code of the subjects, properly encrypted in accordance with privacy regulations.

### 2.2. Adherence Measurement

Medication adherence was estimated according to the EMERGE guidelines [18], focusing on two adherence phases: implementation, i.e., the extent to which a patient’s actual dosing corresponds to the prescribed dosing regimen, from initiation until the last dose, and discontinuation, i.e., the moment the patient stops taking the prescribed medication, which includes persistence, i.e., the length of time between initiation and the last dose, which immediately precedes discontinuation. Persistence was evaluated by estimating switching rates intended as the refill of any other HF drug treatment, namely diuretics (ATC II: C03), beta-blocking agents (ATC II: C07), and agents acting on the renin-angiotensin system (ATC II: C09). Trajectory-based modeling, based on cluster analysis, was used to evaluate both phase implementation and discontinuation. Clustering is a set of multivariate data analysis techniques aimed at selecting and grouping homogeneous items in a data set [33,34,35,36]. Cluster analysis for medication adherence measurement allowed the identification of groups of subjects (namely, subjects with heart failure treated with sac/val) with common characteristics (e.g., potential determinants or predictors of risk) [21,34,35,36]. GBTM analyses were performed with the AdhereR package (version 0.8.10) [19,20]. Refill histories for a single medication (sac/val) over an observation period of 365 days (1-year) were calculated on the identified cohort of HF subjects starting treatment with any dosage of sac/val. According to the approved EMA guidelines regarding the sac/val posology [5], two dose-die were considered to construct the refill history of each subject’s trajectory. After the initial fill, refill durations of 30, 60, or 90 days were randomly sampled for each subsequent refill (sensitivity analyses). A refill duration of 30 days was finally set. The indicator used for the adherence longitudinal assessment was continuous multiple interval measures of medication availability/gaps (CMA) version 9 estimates for the whole observation period [19,20] computed as the ratio of theoretical medication use days to the adherence assessment period duration, accounting for supply carryover and excluding end-period supply. The CMA9 established a consistent average adherence value by weighing days according to individual supply ratios, calculated without overlap in one-month sliding windows, matching the typical delivery period between successive supplies. Clustering in adherence groups was performed with the R package “kml” (version 2.4.1), which provided an implementation of k-means designed to work specifically on longitudinal data [37]. K-means clustering used a random initial state; hence, the optimal solution of cluster number was selected by maximizing the Calinski–Harabasz criterion [37].

### 2.3. Sociodemographic and Clinical Covariates

Drug utilization profiles of subjects belonging to each adherence trajectory were assessed by comparing age, sex, polypharmacy, hospital admissions, type of comedications, type of other comedications as covariates. Patient complexity was evaluated with the age-adjusted Charlson comorbidity (ACCI) index and level of polypharmacy. The ACCI score was calculated for each subject considering all comorbidities, with additional points added for age and dichotomizing patients into three groups: low score (0–1), mild score (2–3), and severe score (≥4) [38]. Polypharmacy was defined according to three classes: “excessive polypharmacy”, intended as the use of ≥10 drugs; “polypharmacy”, as the use of 5 to 9 drugs; and “no-polypharmacy”, as the concomitant use of ≤4 drugs.

### 2.4. Statistical Analysis

A multinomial regression analysis was performed to assess the relationship between demographic and complexity indexes (covariates) and adherence trajectory groups. A stepwise selection procedure was used to select covariates to be included in the models, considering a *p* value < 0.05 statistically significant.

Data management was performed with Microsoft SQL server (version 2018), and all analyses were performed in R (version 3.6, The R Formulation for Statistical Computing, Vienna, Austria).

## 3. Results

### 3.1. Cohort Characteristics

Overall, over the 3-year study period, 4455 HF-incident subjects starting sac/val treatment were identified and included in the analyses (Figure 1). Of those, 70% were male, and more than half were aged between 51 and 75 years (n = 2653, 59.7%). Patient complexity evaluated with the age-adjusted Charlson comorbidity (ACCI) index recorded an overall mean score of 7.9 SD 6.2. Thus, confirming the ACCI score, 37.1% of the cohort was in an excessive polypharmacy regimen, e.g., was treated with more ≥10 drugs per day. Moreover, among subjects with at least one previous hospital admission, 33.4% were hospitalized for cardiovascular causes, and about 20% had more than 2 hospitalizations for different causes. Hence, most cardiovascular comorbidities were cardiomyopathy (n = 422, 9.5%) and chronic coronary syndrome (CCS) (n = 339, 7.6%). The most common HF-related comorbidities were diabetes (n = 506, 11.4%) and chronic kidney disease (n = 206, 4.6%). Overall cohort characterization is shown in Table 1 and Appendix A.

### 3.2. Adherence Measurement

From the longitudinal trajectory analysis, four adherence clusters were selected based on their medication refill histories with overlapping CMA9 trajectories (Figure 2 and Table 2). The four distinct adherence trajectories identified were populated with the following:

Group A: “High adherence” with an average CMA9 of around 95%, including 42.6% of subjects; among those who discontinued treatment, 13.8% switched to other HF medications, and most of them within one month from the sac/val treatment initiation (86.6%);

Group B: “Partial drop-off” with high adherence initially (CMA9 of around 85%) and partial drop after some time (CMA9 of around 10%), including 19.6% of subjects; among those who discontinued treatment, 10.5% switched to other HF medications within six months from the sac/val treatment initiation (70.7%);

Group C: “Moderate-adherence” with a median CMA9 between 50 and 70%, including 19.3% of subjects; among those who discontinued treatment, 23.4% switched to other HF medications within one month from the sac/val treatment initiation (59.9%);

Group D: “Low-adherence” with one or two refills after the initial fill and no refills afterward, including 18.4% of subjects; among those who discontinued treatment, 15.5% switched to other HF medications within one month of the sac/val treatment initiation (41.7%).

All characteristics of subjects grouped according to similar estimates of medication adherence and with similar medication-taking behaviors are detailed in Table 3. The high value of the age-adjusted Charlson comorbidity index score was detected in HF subjects clustered in the low-adherence group (9.9, SD 7.9). Moreover, subjects with the lowest medication adherence had the highest number of previous hospital admissions (1.9, SD 1.3) compared to subjects with the highest adherence levels.

Finally, logistic regression models identified the association between subjects’ characteristics as determinants of whether they belonged to one adherence group or not, as graphically shown in Figure 3a,b. Polypharmacy was associated with partial drop-off adherence (OR 1.194, 95% CI 1.175–1.214), while the occurrence of more than one HF hospitalization (OR 1.165, 95% CI 1.151–1.179) or other hospitalizations (OR 1.481, 95% CI 1.459–1.503) was associated with low adherence (Figure 3a).

Regarding cardiovascular-related comorbidities, subjects suffering from chronic coronary syndrome (CCS) (OR 2.514, CI 95% 2.341–2.701) were more likely to have a moderate adherence to sac/val treatment; particularly, subjects with a NSTEMI and STEMI event (OR 1.529, CI 95% 1.478–1.582 and OR 1.110, CI 95% 1.072–1.149, respectively) and suffering from hypertension (OR 1.34, CI 95% 1.297–1.384) were more likely to have a partial drop-off in adherence after about six months from the treatment initiation; also, having had a STEMI event also proved to be a determinant of low adherence (OR 1.18, CI 95% 1.141–1.221). Regarding other comorbidities that were HF-related, subjects suffering from chronic kidney disease (OR 1.595, CI 95% 1.55–1.641) and diabetes (OR 1.212, CI 95% 1.187–1.236) were more likely to have a moderate adherence to treatment; those suffering from respiratory failure (OR 1.079, CI 95% 1.051–1.107) were more likely to have a low adherence behavior (Figure 3b).

## 4. Discussion

The central finding of this study underscores a critical aspect of managing heart failure (HF) patients prescribed sacubitril/valsartan (sac/val): less than 50% of individuals maintain high medication adherence within one year from their initial prescription. Our study unveils the complexity of adherence behavior among HF patients, delineating four distinct adherence patterns. These findings raise pivotal questions about the role of healthcare systems and professionals in the early identification of patients falling into these diverse adherence categories and how best to provide tailored support to optimize treatment outcomes. The results showed the different characteristics of subjects belonging to diverse adherence groups, confirming the association of determinants of specific adherence behavior. Particularly, the determinants of belonging to a low medication adherence behavior were related to high rates of polypharmacy and multimorbidity regimens as well as the frequency of previous hospitalizations for causes not related to the primary HF condition. Our findings revealed an interesting pattern where subjects belonging to the low-adherence group demonstrated a notably elevated age-adjusted Charlson comorbidity index score, averaging 9.9 (with a standard deviation of 7.9). This observation suggests that individuals with heart failure in the low-adherence group tend to have more complex comorbidities, potentially contributing to their reduced adherence to medication regimens. Additionally, it is worth noting that subjects with the lowest medication adherence levels also exhibited a higher number of prior hospital admissions, averaging 1.9 (with a standard deviation of 1.3), in comparison to subjects with higher adherence levels. This relationship may indicate a link between lower medication adherence and a history of more frequent hospitalizations, suggesting the need for targeted interventions to enhance adherence and reduce hospital readmissions among this patient subgroup.

In addition, results suggested that STEMI events and concomitant hypertension have been shown to be determinants of partial drop-off in adherence within six months of treatment initiation, observing a switch of these subjects to standard HF medications. Specifically, the most challenging clinical scenario was found for HF patients clustered into the low adherence group to (17%), noting in such patients the conditioning STEMI episode. This is understandable, as over the past two decades it has been recognized that STEMI is one of the major risk factors for the occurrence of HF, identifying HF as a complication of MI [39,40,41,42,43,44]. Moreover, in line with our results, it has also been shown that the coexistence of the two diseases complicates the overall clinical pattern [39,40,41,42,43,44], resulting in poor adherence to prescribed therapies with consequent negative clinical outcomes.

Our findings also demonstrate similarities in subjects clustered into the partial drop-off in the adherence group. These subjects, who discontinued treatment within six months of starting sac/val therapy, showed a complex clinical pattern with more than two prior hospitalizations for non-HF-related causes and had hypertension and a NSTEMI episode at the time of HF diagnosis and initiation of sac/val treatment. This can be justified following the results achieved with the PARADIGM-HF trial, in which sac/val was shown to be superior to enalapril in reducing hospitalizations for worsening HF, cardiovascular mortality, and all-cause mortality in subjects with HFrEF with LVEF ≤ 40% [6,7,8,9,45]. Therefore, the choice to start treatment with sac/val is justifiable, but these subjects maintained high levels of adherence only in the first few months of treatment.

Similarities were also identified in subjects clustered within the perfect adherence behavior group. Hence, subjects newly diagnosed with HF being treated concomitantly with antidiabetic drugs maintained significantly high levels of adherence to sac/val treatment (80% in the first year of therapy). Corresponding to this, although it is recognized that diabetes is an independent risk factor for heart failure progression [43], a post hoc analysis from the PARADIGM-HF trial [43] also demonstrated additional benefits of sac/val treatment in the reduction in the incidence of diabetes and glycemic control maintenance. This may explain the better management of concomitant pharmacological therapies to treat the two chronic diseases.

Two recent studies conducted in the United States have already assessed adherence to sacubitril/valsartan treatment, and their findings align with our own results [10,11]. Specifically, they have observed that overall adherence to sacubitril/valsartan is deemed acceptable, yet it tends to diminish in individuals with higher comorbidities or those who initiated therapy after recent hospitalization [10]. Moreover, these studies have highlighted a significant finding: high adherence to sacubitril/valsartan is intricately associated with substantially reduced rates of hospital readmissions and mortality [11]. It is worth noting that these studies, in contrast to our own research, take a holistic approach to evaluating adherence levels without discerning the distinct adherence behaviors that can be identified through the utilization of the group-based trajectory modeling methodology. While these prior studies offer valuable insights into adherence patterns and clinical outcomes, our study contributes by applying a more refined analysis and differentiating and characterizing diverse adherence behaviors through clusterization using the group-based trajectory modeling approach. This approach enables a more granular and precise understanding of adherence dynamics, ultimately offering a comprehensive view of how adherence patterns impact clinical outcomes in a way that broader assessments might not capture.

Our study is subject to several limitations. First, administrative/electronic healthcare databases do not contain prescription information relevant for adherence measurement, such as the reasons behind sac/val discontinuation or the consequences of discontinuation. For instance, a clinically justifiable reason for discontinuation of sac/val is identifiable in symptomatic hypotension, which was reported more commonly in subjects treated with sac/val as compared to enalapril, but despite developing hypotension, these subjects also gained clinical benefits from sac/val [7,8,9]. Hence, as recommended from the 2021 ESC Guidelines for the diagnosis and treatment of acute and chronic heart failure, sac/val is recommended as a replacement for an ACE-inhibitor in patients with HFrEF to reduce the risk of HF hospitalization and death [46]. Second, there is no information available on free samples distributed after an HF hospitalization, which may cause some initial months of therapy to be missed. Third, study results are not generalizable to subjects who died with HF during follow-up because they were excluded from the analyses, as group-based trajectory models cannot handle random, non-missing data. Fourth, a limitation of our study pertains to the inherent treatment complexity associated with heart failure patients, who typically receive therapy involving a minimum of three to four distinct pharmacological classes, as per European clinical guidelines. Consequently, we classified patients as being on polypharmacy only if they exceeded five unique medication prescriptions.

On the other hand, this study has several strengths. The major is surely related to the study methods based on longitudinal and dynamic calculation of adherence to sac/val. Based on the results hitherto observed in this study, medication adherence can be assessed using longitudinal data to avoid returning a dichotomous value of adherence/non-adherence. It is also based on the use of clustering of subjects with common characteristics (determinants/predictors of an adherence level). The reason why clustering on longitudinal adherence trajectories was performed is that it offers advantages over simple clustering on groups, means distinct longitudinal adherence patterns, and allows classification accuracy for different scenarios [37].

Thus, clustering HF patients according to their medication-taking behavior with similarities in clinical and baseline characteristics can help clinicians identify upstream patients’ most at-risk clinical inefficacy. Identifying individuals at risk of non-adherence early in the treatment process is of paramount importance. Early recognition enables healthcare providers to implement tailored interventions aimed at improving patient adherence, which, in turn, may enhance overall health outcomes and reduce healthcare utilization, including costly hospital readmissions. These interventions might involve personalized education, medication management tools, and close monitoring of high-risk patients. By addressing non-adherence proactively, we can work towards optimizing patient care and potentially alleviating the burden on healthcare resources.

## 5. Conclusions

The study’s identification of four distinct adherence patterns among patients with heart failure (HF)-prescribed sacubitril/valsartan (sac/val) carries significant implications for healthcare practice, research, and policymaking, both locally and globally. Recognizing the potential for a substantial proportion of HF patients to achieve high adherence levels opens avenues for clinicians to build upon this success through education and monitoring. Equally crucial is addressing the complexities observed in other adherence groups, necessitating early identification and tailored interventions. The group-based trajectory modeling (GBTM) methodology employed in this study proves invaluable in understanding dynamic adherence behaviors, offering researchers a powerful tool to inform evidence-based interventions for HF and chronic conditions more broadly. Policymakers can leverage these insights to formulate targeted guidelines, optimize resource allocation, and enhance health outcomes.

## Figures and Tables

**Figure 1 pharmaceutics-15-02568-f001:**
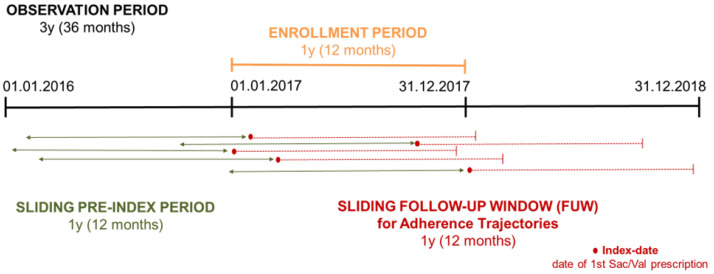
Overview of the study period.

**Figure 2 pharmaceutics-15-02568-f002:**
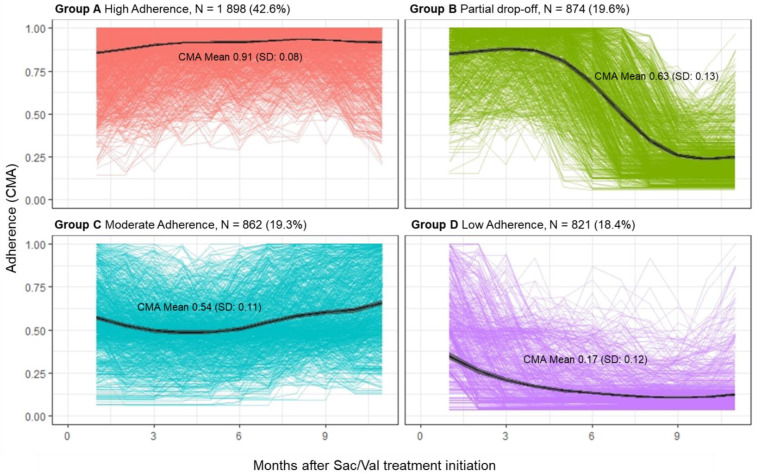
Longitudinal adherence trajectories of sac/val incident subjects.

**Figure 3 pharmaceutics-15-02568-f003:**
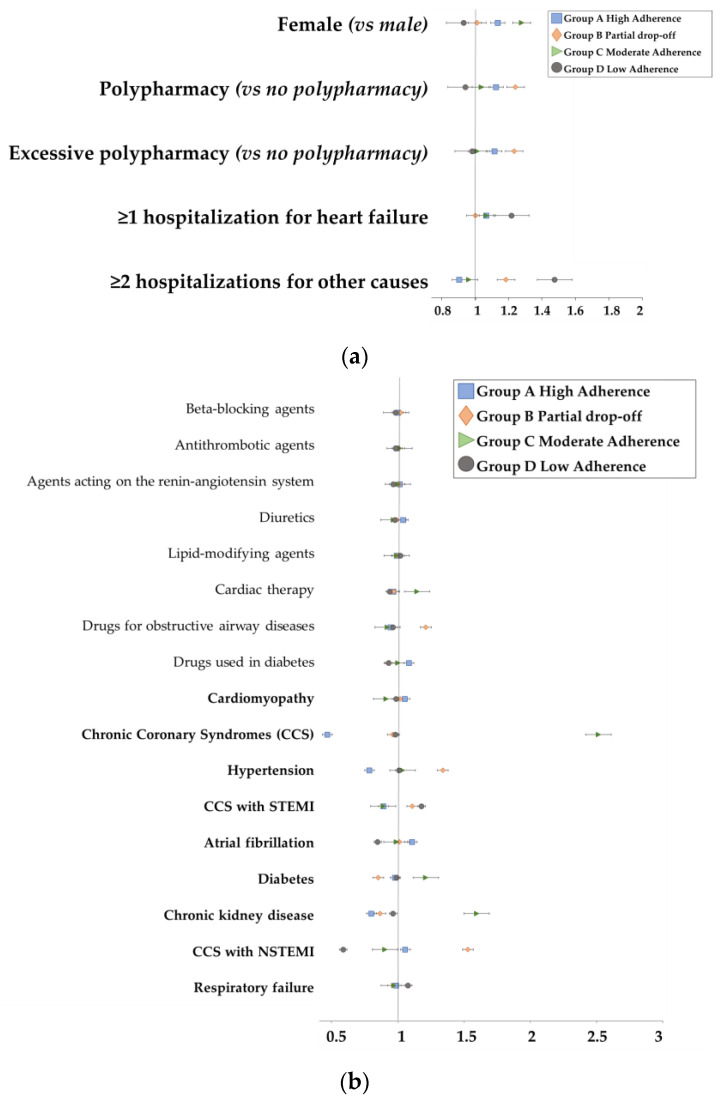
(**a**). Logistic regression models of baseline characteristics as predictors of adherence trajectories. (**b**). Logistic regression models of comedications and comorbidities (in bold) as predictors of adherence trajectories.

**Table 1 pharmaceutics-15-02568-t001:** Baseline cohort’s characteristics.

Characteristics	Overall N = 4455
Gender, N (%)	
Female	1336 (30.0)
Male	3119 (70.0)
Age, Mean (SD)	69.1 ± 12.0
Age groups, N (%)	
0–25 y	17 (0.4)
26–50 y	306 (6.9)
51–75 y	2653 (59.7)
over 76 y	1469 (33)
Index Dosage, N (%)	
Low dosage (24 mg/26 mg)	2941 (66)
Medium dosage (49 mg/51 mg)	1306 (29.3)
High dosage (97 mg/103 mg)	208 (4.7)
Polypharmacy, * N (%)	
No polypharmacy (1–4 drugs)	1363 (30.6)
Polypharmacy (5–9 drugs)	1410 (31.6)
Excessive polypharmacy (≥10 drugs)	1654 (37.1)
ACCI score, Mean (SD) **	7.9 (6.2)
Hospital admission, * Mean (SD)	1.7 (1.1)
Had ≥1 hospitalization for HF, N (%)	1489 (33.4)
Had ≥2 hospitalizations for other causes, N (%)	875 (19.6)
Medications HF-related, * N (%)	
Beta-blocking agents	4080 (91.6)
Antithrombotic agents	3900 (87.6)
Agents acting on the renin-angiotensin system	3790 (85.1)
Diuretics	3689 (82.8)
Other medications, * N (%)	
Lipid-modifying agents	3155 (70.8)
Cardiac therapy	2471 (55.5)
Drugs for obstructive airway diseases	1977 (44.4)
Drugs used in diabetes	1655 (37.2)
Cardiovascular comorbidities, * N (%)	
Cardiomyopathy	422 (9.5)
CCS	339 (7.6)
Hypertension	221 (5)
CCS with STEMI	179 (4)
CCS with NSTEMI	164 (3.7)
Atrial fibrillation	149 (3.3)
Other comorbidities, * N (%)	
Diabetes	506 (11.4)
Chronic kidney disease	206 (4.6)
Respiratory failure	132 (3)

* Conditions occurring one year prior to the index date of initiation of sac/val therapy. ** Index calculated on hospitalized subjects (n = 2900). Notes: low dose: sacubitril 24 mg/valsartan 26 mg; medium dose: sacubitril 49 mg/valsartan 51 mg; and high dose: sacubitril 97 mg/valsartan 103 mg. Abbreviations: CCS, Chronic coronary syndrome; HF, Heart failure; NSTEMI, non-ST elevation myocardial infarction; SD, Standard deviation; and STEMI, ST elevation myocardial infarction.

**Table 2 pharmaceutics-15-02568-t002:** Adherence profiles of sac/val incident subjects.

Patients’ Adherence Profiles	Group A High Adherence	Group B Partial Drop-Off *	Group C Moderate Adherence	Group D Low Adherence
	N = 1898	N = 874	N = 862	N = 821
CMA, Mean (SD)	0.91 (0.08)	0.63 (0.13)	0.54 (0.11)	0.17 (0.12)
Days on treatment, Median (IQR)	322 (103)	173.5 (93.5)	157 (89)	79.5 (57)
Switchers, ° N (%)	261 (13.8)	92 (10.5)	202 (23.4)	127 (15.5)
Switch after 1 month from index date ^§^	226 (86.6)	20 (21.7)	121 (59.9)	53 (41.7)
Switch after 2 months from index date ^§^	28 (10.7)	7 (7.6)	58 (28.7)	40 (31.5)
Switch after 6 months from index date ^§^	7 (2.7)	65 (70.7)	23 (11.4)	34 (26.8)

* Partial drop-off: high adherence initially and partial drop after some time. ° Patients switched to any heart failure drug treatment: diuretics (ATC II: C03); beta-blocking agents (ATC II: C07); and agents acting on the renin-angiotensin system (ATC II: C09). ^§^ Percentage computed from total switchers. Abbreviations: IQR, interquartile range.

**Table 3 pharmaceutics-15-02568-t003:** Characteristics of sac/val subjects via adherence trajectory.

	Group A High Adherence	Group B Partial Drop-Off	Group C Moderate Adherence	Group D Low Adherence	*p*-Value
Total, ° N (%)	1898 (42.6)	874 (19.6)	862 (19.3)	821 (18.4)	
Age, Mean (SD)	69.0 (11.3)	69.1 (11.8)	69.3 (11.7)	69.2 (13.8)	0.003
Sex, N (%)					0.003
Female	549 (28.9)	270 (30.9)	228 (26.5)	289 (35.2)	
Male	1349 (71.1)	604 (69.1)	634 (73.5)	532 (64.8)	
Polypharmacy, * N (%)					0.001
No polypharmacy (1–4 drugs)	561 (29.6)	248 (28.4)	270 (31.3)	284 (34.6)	
Polypharmacy (5–9 drugs)	617 (32.5)	269 (30.8)	291 (33.8)	233 (28.4)	
Excessive polypharmacy (≥10 drugs)	715 (37.7)	351 (40.2)	298 (34.6)	290 (35.3)	
ACCI score, Mean (SD) **	7.5 (5.5)	7.6 (5.9)	8.2 (6.1)	9.0 (7.9)	0.001
Hospital admission, * Mean (SD)	1.6 (1.0)	1.7 (1.1)	1.7 (1.1)	1.9 (1.3)	0.001
≥1 hospitalization for HF, N (%)	637 (33.6)	278 (31.8)	293 (34)	281 (34.2)	0.001
≥2 hospitalizations for other causes, N (%)	931 (49.1)	426 (48.7)	179 (20.8)	185 (22.5)	0.005
Medications HF-related, * N (%)					
Beta-blocking agents	1682 (88.6)	770 (88.1)	767 (89)	686 (83.6)	<0.001
Antithrombotic agents	1599 (84.2)	740 (84.7)	732 (84.9)	662 (80.6)	<0.001
Agents acting on the renin-angiotensin system	1587 (83.6)	737 (84.3)	701 (81.3)	600 (73.1)	<0.001
Diuretics	1538 (81)	692 (79.2)	687 (79.7)	614 (74.8)	<0.001
Other medications, * N (%)					
Lipid-modifying agents	1335 (70.3)	599 (68.5)	572 (66.4)	512 (62.4)	<0.001
Cardiac therapy	1021 (53.8)	436 (49.9)	501 (58.1)	406 (49.5)	<0.001
Drugs for obstructive airway diseases	788 (41.5)	349 (39.9)	369 (42.8)	386 (47)	<0.001
Drugs used in diabetes	689 (36.3)	321 (36.7)	275 (31.9)	298 (36.3)	<0.001
Cardiovascular comorbidities, * N (%)					
Cardiomyopathy	189 (10)	70 (8)	90 (10.4)	73 (8.9)	<0.001
CCS	157 (8.3)	62 (7.1)	70 (8.1)	50 (6.1)	<0.001
Hypertension	79 (4.2)	47 (5.4)	43 (5)	52 (6.3)	0.001
CCS with NSTEMI	69 (3.6)	42 (4.8)	28 (3.2)	25 (3)	0.001
CCS with STEMI	66 (3.5)	37 (4.2)	36 (4.2)	40 (4.9)	<0.001
Atrial fibrillation	65 (3.4)	22 (2.5)	30 (3.5)	32 (3.9)	<0.001
Other comorbidities, * N (%)					
Diabetes	203 (10.7)	90 (10.3)	105 (12.2)	108 (13.2)	0.001
Chronic kidney disease	74 (3.9)	41 (4.7)	48 (5.6)	43 (5.2)	0.002
Respiratory failure	45 (2.4)	28 (3.2)	25 (2.9)	34 (4.1)	<0.001

° Percentage calculated on the total of subjects analyzed. * Conditions occurring one year prior to the index date of initiation of sac/val therapy. ** Index calculated on hospitalized subjects (n = 2900). Abbreviations: ACCI, Age-Adjusted Charlson Comorbidity Index score; CCS, Chronic coronary syndrome; HF, Heart failure; NSTEMI, non-ST elevation myocardial infarction; SD, Standard deviation; and STEMI, ST elevation myocardial infarction. Notes: *p*-value less than 0.05 was considered statistically significant.

## Data Availability

All data used for the current study are available upon reasonable request to the Centro di Ricerca in Farmacoeconomia e Farmacoutilizzazione (CIRFF) authorized by the governance board of Unità del Farmaco della Regione Campania [D.G.R. 276, 23 May 2017].

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
