# Peer review of "Longitudinal Trajectory Modeling to Assess Adherence to Sacubitril/Valsartan among Patients with Heart Failure"

_pharmaceutics, 2023, doi:10.3390/pharmaceutics15112568_

Round 1

Reviewer 1 Report

Comments and Suggestions for Authors

This paper aims to assess the adherence to sacubitril/valsartan among patients with heart failure through a longitudinal trajectory modeling.

The article shows a serious issue of important clinical relevance.

Overall, the work is interesting and well written.

However, I would suggest to the authors to expand some aspects:

-        Line 85 and 86 mention inclusion and exclusion criteria, but they are not described in the manuscript. Please explain them in the methods section.

-        There is clear confusion regarding cardiovascular-related comorbidities. Line 226 distinguishes patients with previous ischaemic heart disease from patients with previous myocardial infarction. “Previous ischaemic heart disease” does not exist. All patients who have had a previous myocardial infarction fall into the category of chronic coronary syndrome, or more commonly known as chronic ischaemic heart disease (https://doi.org/10.1093/eurheartj/ehz425). Since 'previous ischaemic heart disease' is not a known nosographic definition, i would eliminate the analyses that took this into account, unless the authors can give a different definition, e.g. patients with previous coronary revascularization in the absence of a previous myocardial infarction.

-        Are there other studies that have evaluated adherence to sal/val? If so, what are the differences compared to your study?

-        Line 292 delete “refs” into the parenthesis

Author Response

Response to Reviewer 1 Comments

1. Summary

Thank you very much for taking the time to review this manuscript. Please find the detailed responses below and the corresponding revisions/corrections in track changes in the re-submitted files.

2. Point-by-point response to Comments and Suggestions for Authors

Comments 1: This paper aims to assess the adherence to sacubitril/valsartan among patients with heart failure through a longitudinal trajectory modeling. The article shows a serious issue of important clinical relevance. Overall, the work is interesting and well written. However, I would suggest to the authors to expand some aspects.

Response 1: Thank you very much for your insightful feedback and recommendations. We greatly appreciate your guidance in enhancing our paper. We have worked to address your suggestions. Methodological aspects have been clarified and refined, and for clinical relevance, we have consulted with clinicians and conducted a comprehensive review of our methodology, ensuring accuracy and relevance. You will find all these corrections and improvements implemented in the revised manuscript, tracked for your convenience.

Comments 2:  Line 85 and 86 mention inclusion and exclusion criteria, but they are not described in the manuscript. Please explain them in the methods section.

Response 2: We have taken measures to provide a more detailed specification of the criteria employed for the final study cohort selection. To facilitate this, we have incorporated an additional figure (study flow chart) that illustrates the selection process for the study cohort. It has been appropriately cited in the text to enhance the clarity of the study methodology.

Comments 3: There is clear confusion regarding cardiovascular-related comorbidities. Line 226 distinguishes patients with previous ischaemic heart disease from patients with previous myocardial infarction. “Previous ischaemic heart disease” does not exist. All patients who have had a previous myocardial infarction fall into the category of chronic coronary syndrome, or more commonly known as chronic ischaemic heart disease (https://doi.org/10.1093/eurheartj/ehz425). Since 'previous ischaemic heart disease' is not a known nosographic definition, i would eliminate the analyses that took this into account, unless the authors can give a different definition, e.g. patients with previous coronary revascularization in the absence of a previous myocardial infarction.

Response 3: Dear Reviewer, we sincerely appreciate your clinical observation concerning the identification of cardiovascular comorbidities in our study. We carefully reviewed the entire methodology of the research and addressed the renaming of comorbidities as suggested. Regarding the term "previous ischaemic heart disease," we acknowledge that this terminology was not consistent with established nosographic definitions. We thoroughly revised the relevant analyses to rectify this inconsistency. Specifically, we replaced the imprecise term with ICD-9-CM codes related to coronary revascularization procedures, aligning it with a more accurate clinical description. In reference to "myocardial infarction" and "subendocardial infarction," we also revisited our methodology for comorbidity identification. The ICD-9-CM codes utilized for these conditions specifically correspond to STEMI and NSTEMI episodes, respectively. Consequently, we have made the necessary adjustments to the analyses to ensure accuracy and precision. We greatly appreciate your guidance, and we would like to provide you with the reference codes used in our methodology for your reference:

·        Coronary Revascularization without Myocardial Infarction: Common procedure codes as Coronary artery bypass graft (CABG) (ICD-9-CM codes 36.10 - 36.19); Percutaneous coronary intervention (PCI) with angioplasty (ICD-9-CM codes 00.66, 00.75, 36.01 - 36.09); and other revascularization procedures.

·        ST Elevation Myocardial Infarction (STEMI): ICD-9-CM code 410.xx, excluding 410.7x

·        Non-ST Elevation Myocardial Infarction (NSTEMI): ICD-9-CM code 410.7x

Comments 4: Are there other studies that have evaluated adherence to sal/val? If so, what are the differences compared to your study?

Response 4: Two additional studies, both conducted in the United States in 2022 (ref. 10) and 2021 (ref. 11), have assessed adherence to Sac/val treatment, yielding findings consistent with the adherence behavior observed in the Italian population in our study. The primary distinctions between our research and these studies lie in the evaluation of adherence levels. We have addressed and discussed these distinctions in the Discussion section.

Comments 5: Line 292 delete “refs” into the parenthesis

Response 5: We have rectified the identified typographical error and conducted a comprehensive revision of the manuscript to ensure overall correctness and clarity. We appreciate your valuable and constructive suggestions.

Reviewer 2 Report

Comments and Suggestions for Authors

Congratulations on a relevant and interesting article, on a topic often forgotten or poorly measured. The conclusions are interesting and, in the current “phenotypic” view of heart failure, they add very clear patient profiles in relation to their therapeutic adherence.

I think I can recommend a series of changes, especially related to clinical concepts, that can improve or make the article more understandable for clinicians, like me, interested in the topic:

-          I understand that the classification according to the number of drugs is based on previous studies, categorized in a similar way. However, in the group of patients with heart failure in whom sacubitril/valsartan is started, there are 4 pharmacological groups with a strong recommendation, so putting the limit of the lower group at 4, I believe, would not adequately describe “polypharmacy” in this group. I think this note should have a section on the limitations of the study, or be modified.

-          In results, in 3.1, I would change HF-related with cardiovascular. Diabetes or kidney disease are “HF-related” problems so they cannot be described as non-HR-related. However, it could be said that they are non-cardiovascular comorbidities.

-          In the classification of comorbidities I detect several serious problems. First, differentiate Ischemic heart disease, Myocardial infarction and subendocardial infarction. The reason is that ischemic heart disease encompasses myocardial infarction, and subendocardial infarction is in turn a type of myocardial infarction. Furthermore, we cannot refer to it as “subendocardial infarction” because we would be talking about a poorly classified pathology. Therefore, I recommend removing this last group, or keeping it under the name “non-ST elevation Myocardial Infarction (NSTEMI)” as the clinical guidelines do. If you do not want to modify this classification and change it to one that is more operational from a clinical point of view, I recommend adding a section on limitations, as it may lead to error or confusion. - Figures 3rd and 3b should be larger to be able to be read clearly.

-          I recommend reviewing, on page 10, scripts 292 to 294, the phrase that begins with “Therefore” and ends with “above” since it is not easily understandable.

-          I would like the patient profiles according to adherence to be commented more extensively in the discussion since it is the real scientific advance that this paper provides.

-          I would like to add, in the discussion or in the conclusion, a summary figure that brings together the conclusions stated on page 11, indents 315 to 320.

-          I would add a point about “use of artificial intelligence” in the acknowledgments if appropriate.

-          In addition, I would like to recommend adding a diagram with the flow of patients, with those who lose follow-up or are excluded for any reason and those finally analyzed.

Author Response

Response to Reviewer 2 Comments

1. Summary

Thank you very much for taking the time to review this manuscript. Please find the detailed responses below and the corresponding revisions/corrections in track changes in the re-submitted files.

2. Point-by-point response to Comments and Suggestions for Authors

Comments 1: Congratulations on a relevant and interesting article, on a topic often forgotten or poorly measured. The conclusions are interesting and, in the current “phenotypic” view of heart failure, they add very clear patient profiles in relation to their therapeutic adherence.

Response 1: Dear Reviewer, on behalf of all the co-authors, we extend our sincere gratitude for your comments and valuable suggestions for improvement. Your thoughtful inputs are addressed in the revised version of manuscript.

Comments 2: I think I can recommend a series of changes, especially related to clinical concepts, that can improve or make the article more understandable for clinicians, like me, interested in the topic.

Response 2: We appreciate your feedback and your perspective as a clinician. Hence, we have made the necessary changes to enhance the clarity and relevance of the text, specifically in relation to the clinical aspects you highlighted. We believe these adjustments will have a positive impact on clinicians' understanding of our topic.

Comments 3: I understand that the classification according to the number of drugs is based on previous studies, categorized in a similar way. However, in the group of patients with heart failure in whom sacubitril/valsartan is started, there are 4 pharmacological groups with a strong recommendation, so putting the limit of the lower group at 4, I believe, would not adequately describe “polypharmacy” in this group. I think this note should have a section on the limitations of the study, or be modified.

Response 3: We appreciate your consideration and agree with your suggestion. As a result, we have added a specific statement within the study's limitations.

Comments 4: In results, in 3.1, I would change HF-related with cardiovascular. Diabetes or kidney disease are “HF-related” problems so they cannot be described as non-HR-related. However, it could be said that they are non-cardiovascular comorbidities.

Response 4: We agree with your correction. We have made the necessary adjustments in the results section of the text to address this inconsistency, defining 'cardiovascular comorbidities' as encompassing all CV-related conditions while distinguishing others as separate comorbidities, and defining Diabetes or kidney disease as "HF-related," as suggested.

Comments 5: In the classification of comorbidities I detect several serious problems. First, differentiate Ischemic heart disease, Myocardial infarction and subendocardial infarction. The reason is that ischemic heart disease encompasses myocardial infarction, and subendocardial infarction is in turn a type of myocardial infarction. Furthermore, we cannot refer to it as “subendocardial infarction” because we would be talking about a poorly classified pathology. Therefore, I recommend removing this last group, or keeping it under the name “non-ST elevation Myocardial Infarction (NSTEMI)” as the clinical guidelines do. If you do not want to modify this classification and change it to one that is more operational from a clinical point of view, I recommend adding a section on limitations, as it may lead to error or confusion. - Figures 3rd and 3b should be larger to be able to be read clearly.

Response 5: Dear Reviewer, we sincerely appreciate your clinical observation concerning the identification of cardiovascular comorbidities in our study. We carefully reviewed the entire methodology of the research and addressed the renaming of comorbidities as suggested. Regarding the term "previous ischaemic heart disease," we acknowledge that this terminology was not consistent with established nosographic definitions. We thoroughly revised the relevant analyses to rectify this inconsistency. Specifically, we replaced the imprecise term with ICD-9-CM codes related to coronary revascularization procedures, aligning it with a more accurate clinical description. In reference to "myocardial infarction" and "subendocardial infarction," we also revisited our methodology for comorbidity identification. The ICD-9-CM codes utilized for these conditions specifically correspond to STEMI and NSTEMI episodes, respectively. Consequently, we have made the necessary adjustments to the analyses to ensure accuracy and precision. We greatly appreciate your guidance, and we would like to provide you with the reference codes used in our methodology for your reference:

·        Coronary Revascularization without Myocardial Infarction: Common procedure codes as Coronary artery bypass graft (CABG) (ICD-9-CM codes 36.10 - 36.19); Percutaneous coronary intervention (PCI) with angioplasty (ICD-9-CM codes 00.66, 00.75, 36.01 - 36.09); and other revascularization procedures.

·        ST Elevation Myocardial Infarction (STEMI): ICD-9-CM code 410.xx, excluding 410.7x

Non-ST Elevation Myocardial Infarction (NSTEMI): ICD-9-CM code 410.7x
Additionally, we replaced Figures 3a and 3b with more larger ones with improved resolution.

Comments 6: I recommend reviewing, on page 10, scripts 292 to 294, the phrase that begins with “Therefore” and ends with “above” since it is not easily understandable.

Response 6: Thank you for the suggestion. We have modified the sentence as follows: “(..) a clinically justifiable reason for discontinuation of sac/val is identifiable in symptomatic hypotension, which was reported more commonly in subjects treated with sac/val as compared to enalapril, but despite developing hypotension, these subjects also gained clinical benefits from sac/val [7–9]. Hence, as recommended from the 2021 ESC Guidelines for the diagnosis and treatment of acute and chronic heart failure, sac/val is recommended as a replacement for an ACE-inhibitor in patients with HFrEF to reduce the risk of HF hospitalization and death [46].”

Comments 7: I would like the patient profiles according to adherence to be commented more extensively in the discussion since it is the real scientific advance that this paper provides.

Response 7: We acknowledge your observation. In fact, we have expanded upon these results in the discussions, providing a more detailed account of the distinctions in patient profiling among the four adherence clusters in the context of heart failure.

Comments 8: I would like to add, in the discussion or in the conclusion, a summary figure that brings together the conclusions stated on page 11, indents 315 to 320.

Response 8: We have provided a summary of the conclusions regarding the main findings of our paper at the end of the discussion section.

Comments 9: I would add a point about “use of artificial intelligence” in the acknowledgments if appropriate.

Response 9: We added in the acknowledgements as follows: “We wish to acknowledge the invaluable role of the programming language R in our research. R, known for its powerful packages for Machine Learning, its ability to handle large datasets, and its open-source nature, has been instrumental in our application of artificial intelligence methodologies. We also extend our appreciation to AdhereR package for its role in facilitating our AI-driven data analysis.”

Comments 10: In addition, I would like to recommend adding a diagram with the flow of patients, with those who lose follow-up or are excluded for any reason and those finally analyzed.

Response 10: We acknowledge your request, and in response, we have included an additional figure in the Supplementary material. This new figure is a Flow chart that illustrates the selection process for the study cohort. It has been appropriately cited in the text to enhance the clarity of the study methodology.

Round 2

Reviewer 1 Report

Comments and Suggestions for Authors

I thank the authors for their efforts to respond to my comments. However, I advise the authors to remove the endpoint "coronary revascularization without myocardial infarction". With the searches done through the ICD codes, it is almost certain that there are patients with previous myocardial infarction. I would rename the endpoint "chronic coronary syndrome", and consider patients with previous NSTEMI and STEMI as subgroups of this category (sac/val prescription occurs at least 3 months after the acute event, so these patients fall under chronic coronary syndromes). 

Author Response

Response to Reviewer 1 Comments

Round 2

Point-by-point response to Comments and Suggestions for Authors

Comments 1: I thank the authors for their efforts to respond to my comments. However, I advise the authors to remove the endpoint "coronary revascularization without myocardial infarction". With the searches done through the ICD codes, it is almost certain that there are patients with previous myocardial infarction. I would rename the endpoint "chronic coronary syndrome", and consider patients with previous NSTEMI and STEMI as subgroups of this category (sac/val prescription occurs at least 3 months after the acute event, so these patients fall under chronic coronary syndromes).

Response 1: We greatly appreciate your valuable comments and suggestions that have allowed us to enhance the quality of our paper. In response to your feedback, we have made the following refinement by explicitly defining the endpoint you suggested as "chronic coronary syndrome (CCS)." Furthermore, we have clarified that episodes of STEMI and NSTEMI are associated with diagnoses of CCS, categorizing them as subgroups (CCS with STEMI and CCS with NSTEMI, respectively). These changes have been implemented both in the manuscript text and in the figures, tables, and supplementary materials. We are truly grateful for the significant support you have provided to our paper.
